# Review on Rice Husk Biochar as an Adsorbent for Soil and Water Remediation

**DOI:** 10.3390/plants12071524

**Published:** 2023-03-31

**Authors:** Zheyong Li, Zhiwei Zheng, Hongcheng Li, Dong Xu, Xing Li, Luojing Xiang, Shuxin Tu

**Affiliations:** 1Hubei Provincial Academy of Eco-Environmental Sciences, Wuhan 430072, China; whityly@163.com (Z.L.);; 2State Key Laboratory of Soil Health Diagnosis and Green Remediation for Environmental Protection, Wuhan 430072, China; 3College of Resources and Environment, Huazhong Agricultural University, Wuhan 430070, China; 4Hubei Research Centre for Environment Pollution and Remediation, Wuhan 430070, China

**Keywords:** rice husk biochar, adsorption, remediation, heavy metals, organic pollutants

## Abstract

Rice husk biochar (RHB) is a low-cost and renewable resource that has been found to be highly effective for the remediation of water and soil environments. Its yield, structure, composition, and physicochemical properties can be modified by changing the parameters of the preparation process, such as the heating rate, pyrolysis temperature, and carrier gas flow rate. Additionally, its specific surface area and functional groups can be modified through physical, chemical, and biological means. Compared to biochar from other feedstocks, RHB performs poorly in solutions with coexisting metal, but can be modified for improved adsorption. In contaminated soils, RHB has been found to be effective in adsorbing heavy metals and organic matter, as well as reducing pollutant availability and enhancing crop growth by regulating soil properties and releasing beneficial elements. However, its effectiveness in complex environments remains uncertain, and further research is needed to fully understand its mechanisms and effectiveness in environmental remediation.

## 1. Introduction

Heavy metals are regarded extremely dangerous environmental pollutants because of their high toxicity, carcinogenicity, and nondegradability [1]. Moreover, most heavy metals have different migration and distribution characteristics—for example, Cd has higher mobility than those of Pb and As—making heavy metal remediation a challenge [2]. At present, technologies such as adsorption, chemical precipitation, membrane removal, ion exchange, complexation, redox, and phytoremediation have been used for heavy metal remediation [3]. However, these techniques are not only expensive but also create secondary pollution [4]. Therefore, there exists an urgent demand for cost-effective, environment-friendly methods for effectively removing heavy metals. In this regard, biochar has attracted widespread attention owing to its high surface area and well-developed pore structure as well as its low cost. Biochar is a carbon-rich porous solid residue produced by the thermal transformation of biomass in partial or total absence of oxygen [5,6]. Biochar production process has three stages: predecomposition (i.e., evaporation of moisture and light volatiles); main pyrolysis (i.e., decomposition of cellulose and hemicellulose), and the formation of carbonaceous solids (i.e., degradation of lignin and other organic matter with strong chemical bonds) [7]. Compared with activated carbon, biochar is a better adsorbent material, with the advantages of low cost, widely available sources, and high affinity for removing heavy metals from contaminated aqueous media [8,9]. Compared with fly ash, biochar has the advantages of green technology, high removal efficiency, and high social acceptability [10]. Compared with clay minerals, biochar has the merits of low cost, high stability, and high affinity for heavy metals [11]. However, biochar has some disadvantages that deserve our attention. Biochar may produce environmentally harmful substances such as polycyclic aromatic hydrocarbons (PAHs) during pyrolysis, and these substances may have a negative impact on the environment. Moreover, applying biochar to the soil may have negative effects on the soil environment; for example, it may cause loosening of the soil and acceleration of soil erosion [12]. Furthermore, previous studies showed that the remediation of heavy metals by biochar can be improved in various ways, such as by changing the pyrolysis conditions, biochar modification with modifiers, and impregnation of biochar with chelating agents [13,14]. A study reported the remediation of heavy metals by biochar in combination with other technologies, such as biochar phytoremediation and biochar electrodynamic remediation [15].

Global rice consumption was approximately 486 million metric tons between 2018 and 2019 [16]. The rice husks produced as a byproduct of this consumption are estimated to be 20%–34% of the amount consumed [17]. Rice husks have a hard surface, low bulk density, and the highest amorphous silica content among grasses [18]. They are primarily composed of cellulose (50%), lignin (30%), and organic compounds (20%) [19]. Despite the potential applications of rice husks as value-added products, most rice producers currently dispose of them through piling and open burning. These disposal methods not only consume land resources but also cause environmental and health problems [20]. Therefore, safe disposing of the rice husk biomass is highly desired. Meanwhile, the high carbon content of rice husk facilitates its conversion into energy-rich biochar upon thermochemical treatment. As one of the many types of biochar, RHB is of great interest because it comes from waste rice husk and is both low-cost and green [21]. RHB has excellent properties and a wide range of applications in several fields. As a precursor resource, RHB is abundant in silicon and alumina, which are suitable for the synthesis of zeolites, amorphous silica and silica-based catalysts. Moreover, its organic components are suitable for preparing activated carbon, carbon-based catalysts and porous carbon [22,23]. RHB is also suitable for soil conditioning. Many studies have focused on the effects of RHB on the chemical properties and biota of soils. Qu et al. [24] showed that RHB can effectively improve the physical and mechanical properties of soils. Further, the surface of RHB contains a large number of acidic (carboxylic, lactonic, and phenolic), neutral (benzoquinonyl and others), and basic functional groups (quinonoid carbonyl, pyrone, and benzopyranyl groups); these functional groups provide the basis for pollution remediation by RHB [25,26]. RHB exhibited good adsorption of heavy metals, organic matter, and other pollutants in water bodies. Notably, it can not only improve the physicochemical properties and reduce the biologically effective heavy metal content in soils but also reduce the heavy metal stress on plants [27]. Additionally, it can be used as a filler material. Recently, many studies showed that rice husk ash can be used as a filler to improve mortar performance drastically. This improvement is attributed to the fact that on addition of rice husk ash, with a smaller particle size than that of the mortar, the compressive strength of mortar is increased because of the filler action [28]. In addition, rice husk ash can promote the sulfate resistance of mortar. Venkatanarayanan and Rangaraju [29] evaluated the effect of rice husk ash on the sulfate resistance of mortars and showed that addition of 15% rice husk ash resulted in a 71% increase in the sulfate resistance of mortar.

Although RHB is widely regarded an environment-friendly adsorbent and soil conditioner, its potential risks cannot be ignored. Table 1 lists the properties and some elemental compositions of RHB. From Table 1, it is seen that RHB contains a variety of harmful components and has abundant silica. Meanwhile, RHB may also release toxic and harmful elements in the environment because of changes in its properties, thereby causing harm to humans and the environment. In recent years, there have been an increasing number of reports on the potential hazards of biochar. These reports mainly addressed the negative impacts of biochar application on soil physicochemical properties, microbiota, greenhouse gases, and organisms at different trophic levels [30,31]. Moreover, Ndirangu et al. [32] discussed the potential risks of biochar prepared from different feedstocks and the necessary mitigation and control measures available. These measures include the selection of suitable feedstocks and control of biochar pyrolysis conditions. Therefore, possible human health hazards and environmental problems caused by rice husk char cannot be ignored; its risk assessment and mitigation control need to be further researched.

Compared to other biochar, RHB is characterized by its widely available sources, high yield, and abundance of amorphous silica [41]. Although numerous review papers on biochar research are available, to the best of our knowledge, no review focusing exclusively on RHB preparation, modification, and purification and remediation of organic and inorganic pollutants in water and soil has been published as yet [42,43]. The present review summarizes the research on the preparation, characterization, and modification of RHB during the decade from 2012 to 2022. Moreover, the purification and remediation effects of RHB applications on water and soil environments, including the improvement of the chemical, biological, and physical properties of soil, plant growth, and stabilization of heavy metals and organic pollutants in soil, are reviewed.

This systematic review used the Web of Science, ScienceDirect, SpringerLink, ACS journal, and Scopus databases. Papers published between 2012 and 2022 were gathered using the search string *rice husk preparation* AND *rice husk biochar modification* AND *rice husk biochar* (*adsorption* OR *sorption*) AND (*water* OR *soil*) AND (*heavy metals*, *organic matter*); only documents published in English were considered.

## 2. Preparation of Rice Husk Biochar

The thermochemical conversion of biomass fuels has three main modes: combustion, gasification, and pyrolysis. The main products of combustion and slow pyrolysis are biochar, and the main products of gasification and fast pyrolysis are biomass oil and biomass gas [44]. Figure 1 summarizes the main modes and products of biomass thermal conversion. From Figure 1 we can see common combustion methods include grate combustion, fluidized bed combustion, and suspension (jet) combustion [18]. Grate combustion produces RHB with a large surface area that is suitable as an adsorbent. Fluidized bed combustion produces RHB that can be used as a filler for polymer composites and ceramic compounds. Suspension (jet) combustion produces RHB with completely amorphous silica, which is suitable for civil engineering and zeolite production [45].

Pyrolysis is popular for its low levels of asset utilization and high recovery rate. It is a complex process wherein the feedstock undergoes thermochemical changes under limited oxygen conditions, producing biochar, biomass oil, and biomass gas. Pyrolysis decomposes the biomass into a mixture of organic and inorganic compounds. Biochar is a carbon-rich, fine-grained, permeable, and aromatically stable solid product [47]. Depending on the operating conditions, pyrolysis can be classified as slow, fast, or flash. Slow pyrolysis is performed at low temperatures, slow heating rates, and long residence times, which facilitates char production. Flash pyrolysis typically has a residence time of less than 0.5 s and a very high heating rate. Fast pyrolysis is performed at moderate temperatures, a high heating rate, and long vapor residence time. Fast pyrolysis and flash pyrolysis favor the formation of bio-oil. Increasing the pyrolysis temperature from 400 °C to 800 °C was shown to decrease the RHB yield from 44.5% to 37.6% [48] and increase the bio-oil yield from 11.26% to 35.92% [49].

## 3. Properties of Rice Husk Biochar

Since rice husk undergoes different chemical reactions under different conditions, the process parameters (mainly pyrolysis temperature, heating rate, and carrier gas flow rate) have a great influence on the yield, structure, composition and properties of RHB.

It has been shown that the pyrolysis temperature is closely related to the changes in the structure and physicochemical properties of RHB. The data about these relationships are shown in Table 2. From the table, it can be seen that: (1) the pH of RHB is positively correlated with the pyrolysis temperature. This is related to the increase in ash content and alkaline functional groups during pyrolysis [50]. (2) In a certain temperature range, the specific surface area and porosity of RHB increase with increasing pyrolysis temperature. This is most likely related to the decomposition of organic matter and the formation of micropores [51]. Paethanom and Yoshikawa [52] observed that pores were created during the pyrolysis of rice husk and these pores led to a sharp increase in the specific surface area of biochar, but this porosity decreased at temperatures above 600 °C. (3) The aromaticity of RHB increased with the increase in pyrolysis temperature [53]. In addition, it has also been shown that high pyrolysis temperatures favor the generation of basic functional groups. This is because as the pyrolysis temperature increases, cellulose and hemicellulose start to degrade and some new functional groups (e.g., carboxyl, lactone, lactone alcohol, quinine, chromene, etc.) are generated. When the pyrolysis temperature is higher (>500 °C), lignin and other hard-to-decompose organic matter begin to decompose, oxygenated acidic functional groups (e.g., carboxyl) are removed in large amounts, and the number of basic functional groups increases [54]. Wei et al. [50] showed that when the pyrolysis temperature increased from 300 °C to 750 °C, the number of basic functional groups increased from 580.16 μmol/g to 1044.29 μmol/g and the number of acidic functional groups decreased from 5431.48 μmol/g to 3841.78 μmol/g. Therefore, RHB produced at lower temperatures (e.g., 300 °C–400 °C) had more organic features and contained more C-O and -OH crown energy groups. RHB produced at higher temperatures (e.g., 600 °C–700 °C) showed highly aromatic carbon layers, larger specific surface area and porosity, which were more favorable for the adsorption of pollutants by RHB.

RHB is often produced at low heating rates [65]. A low heating rate reduces the occurrence of not only side reactions but also thermal cracking reactions in the biomass, thereby increasing the RHB yield. Phuong et al. [66] found that increasing the heating rate from 10 °C/min to 50 °C/min decreased the RHB yield from 44% to 38%. Lower heating rates are also beneficial for obtaining more RHB. A low heating rate promotes the formation of an aromatic structure, which is conducive to obtaining more aromatic functional groups and stabilizes the carbon structure of RHB [67]. Abbas et al. [56] found that increasing the heating rate from 1 °C/min to 10 °C/min decreased the RHB carbon content from 71.05% to 67.46%, increased the oxygen/carbon ratio from 0.018% to 0.067%, and increased the hydrogen/carbon ratio from 0.351% to 0.587%.

The pyrolysis of biomass generates vapors that become involved in secondary reactions if they are not removed in time. This affects the composition of the pyrolysis products and thus the RHB yield. The biochar yield was found to stay the same when the nitrogen flow rate exceeded a certain level [68]. This indicates that lowering the carrier gas flow rate can remove most of the vapors from the reaction zone, thus reducing the occurrence of secondary reactions.

## 4. Modification of Rice Husk Biochar

Virgin RHB generally has a limited adsorption capacity for pollutants. To improve the applicability of RHB to pollutant remediation, it should be modified. The RHB surface is rich in functional groups, which is favorable for modifier loading [5]. Common modification methods for enhancing the adsorption capacity and remediation effectiveness of RHB include physical, chemical, and microbial approaches [69]. Physical modifications are easy to apply because of their simplicity and low cost [70]. Chemical modification can be used to obtain RHB with different modification effects depending on the application, but such modifications may cause secondary contamination and incur high costs. Microbial modification is less expensive and produces no secondary contamination, and the modified RHBs tend to have better pollutant adsorption ability, but such modification often requires a longer time to achieve good modification results [71].

### 4.1. Physical Modification

Physical modification of RHB is applied during the charring and activation processes. The charring process occurs under anaerobic or anoxic conditions; the rice husks undergo pyrolysis at a certain temperature, which opens some of the pores [72]. The activation process mainly occurs in the presence of an activator (e.g., water vapor, carbon dioxide). The activator further opens the pores and increases the specific surface area of RHB [73,74]. Therefore, physical modification mainly involves increasing the specific surface area and porosity of RHB to improve the adsorption capacity of target pollutants. Physical modification is generally performed by steam activation, carbon dioxide activation ultraviolet radiation, and ball milling [75]. Steam activation is the most widely used owing to its simplicity and nonpolluting nature [76]. Mayakaduwa et al. [63] prepared RHB at 700 °C and then applied steam activation for 45 min, which increased the ability of RHB to remove carbofuran (i.e., a toxic insecticide) from an aqueous solution by 2.4 times to 396 mg/g.

### 4.2. Chemical Modification

Several approaches are available for chemical modification, of which several are listed below.

(1)Acid/alkali modification

Acid modification refers to the treatment of rice husks or RHB with acid reagents (e.g., nitric acid, hydrochloric acid, and phosphoric acid) [21,77] to change the physicochemical properties. These changes can be divided into four main categories: removing metal impurities from the RHB surface, introducing acidic functional groups to the RHB surface, applying a corrosive acid to the RHB surface to make it inhomogeneous and increase the porosity as well as specific surface area, and protonating the RHB surface to increase the ion exchange rate between H^+^ and heavy metals [74,78]. Zhao et al. [79] found that pretreating biochar with phosphoric acid significantly increased the specific surface area (from 51.0 to 930 m^2^/g), total pore volume (from 0.046 to 0.558 cm^3^/g), and number of micropores (from 59.0% to 78.4–81.9%).

Alkali modification uses alkaline reagents instead, which generally activate the internal structure of RHB and increase the specific surface area to improve the adsorption and fixation capacities [80]. Some alkali modifiers remove ash impurities and nonpyrolyzed organic matter on the RHB surface, which increases porosity. Other alkali modifiers (e.g., potassium hydroxide) form intercalating compounds with inner and outer layers in RHB, thereby increasing the specific surface area [81,82]. Tsai et al. [83] modified RHB with sodium hydroxide, which increased the specific surface area from 21.764 to 434.62 m^2^/g and improved the adsorption capacity for malachite green by 4.3 times.

(2)Metal (metal oxide) modification

Metal (metal oxide) modification changes the specific surface area, metal element content, cation exchange capacity, and surface charge of RHB. For example, Xiang et al. [84] used magnesium oxide to modify RHB, which increased the specific surface area by 4.88 times to 20.64 m^2^/g. RHB can also be modified by using iron oxide, magnesium, aluminum, titanium dioxide, manganese dioxide, etc. [85]. These modifications are usually performed via impregnation and in situ synthesis. The former involves impregnating RHB with a metal or metal oxide, which promotes the attachment of metal ions. The latter involves the direct addition of metal or metal oxide reagents to the feedstock. Then, pyrolysis, chemical precipitation, and activation are performed to complete the modification [86]. In situ synthesis has two major advantages over impregnation. First, it has more controllability, and improve the stability of the modifier. Second, it improves the adsorption capacity for the target pollutant more than the impregnation method when other conditions such as the raw material and processing conditions are maintained [87]. Teng et al. [27] found that iron modification significantly increased the effectiveness of RHB at solidifying lead and antimony in contaminated soil by eight and five times to 25% and 40%, respectively.

(3)Functional group modification

Functional group modification uses organic compounds with target functional groups that are capable of chemically bonding with the RHB surface, which then loads the target functional group onto the RHB [88]. Functional group modification can significantly increase the number of functional groups in RHB, which provides more adsorption sites for target pollutants [89]. Functional group modification can be divided into several categories depending on the contents of the functional groups: oxygen-containing, nitrogen-containing, sulfur-containing, hydrophobic, and others [90]. Nitrogen-containing functional groups have a strong affinity for complexing with target pollutants such as heavy metals, particularly heavy metal cations such as Cd^2+^, Zn^2+^, and Cu^2+^ [91]. Nitrogen-containing functional groups are usually introduced to the RHB surface through nitrification and are subsequently reduced to the corresponding amino derivatives by reduction [92]. Gai et al. [93] found that using nitrogen-containing functional groups to modify RHB significantly increased the Cu^2+^ adsorption capacity by 2.2 times to 29.11 mg/g.

### 4.3. Microbial Modification

Microbial modification uses microorganisms, which act synergistically with RHB to considerably improve the adsorption capacity and remediation effectiveness of target pollutants [94]. RHB is used to host bacteria, which start to multiply inside the RHB at the ideal temperature and when sufficient nutrients are available to form a film on the RHB surface. The microorganisms use extracellular enzymes to convert some hard-to-degrade organic matter into easy-to-degrade organic matter. They then convert the biodegradable organic matter into biomass, carbon dioxide, and other substances. Finally, RHB removes the above substances through adsorption. Microbial modification improves the physicochemical properties of RHB such as the structure, redox potential, and pH. Cheng et al. [95] used the mycobacterium (*Xylella vulgaris*) to modify RHB, which improved the adsorption capacity for toluene by 31.6%.

## 5. Rice Husk Biochar as an Adsorbent

Water pollution poses a serious threat to the environment and human health because of its rapid spread, mobility, and impact. The effective removal of water pollution has been an issue of public concern for many years. RHB is widely used for the adsorption of heavy metals and organic substances in water because of its good stability, recyclability, and adsorption capacity [96].

### 5.1. Adsorption of Heavy Metals in Water

RHB has five main mechanisms for heavy metal adsorption (Figure 2) [77,97,98]: physical adsorption between RHB and heavy metal ions; ion exchange between ions adsorbed on the RHB surface (e.g., K^+^, Na^+^, Mg^2+^) and heavy metals; complexation of heavy metals with functional groups (e.g., carboxyl, hydroxyl) on the RHB surface; electrostatic interaction between heavy metals and the RHB surface; and coprecipitation between heavy metals and metal salts on the RHB surface (e.g., lead–phosphate–silicate precipitation).

RHB may be able to adsorb more heavy metals in water than biochar produced from other agricultural wastes. Agricultural wastes such as wood, straw, and shells are often used as a raw material for biochar. One ton of agricultural waste is estimated to yield about 0.3 tons of biochar [99,100]. Table 3 compares the differences in the physicochemical properties of biochar prepared from several common types of agricultural waste. Compared with biochar made from corncobs, straw, and sawn wood, RHB has a greater specific surface area and ash content as well as more abundant aromatic functional groups. Sanka et al. [101] showed that RHB has a superior adsorption capacity for chromium, iron, and lead in industrial wastewater to that of corncob biochar. RHB removed 65% of chromium and 90% of lead from industrial wastewater; however, corncob biochar only removed 20% and 35% of chromium and lead, respectively. Amen et al. [102] found that biochar prepared from agricultural waste had high adsorption capacities for Pb^2+^ and Cd^2+^. RHB, wheat straw biochar, and corncob biochar adsorbed 96.41%, 95.38%, and 96.92%, respectively, of Pb^2+^ and 94.73%, 93.68%, and 95.78%, respectively, of Cd^2+^. Higashikawa et al. [103] showed that RHB had a higher removal capacity for Cd^2+^ and Ni^2+^ in solution than sawdust biochar. Thus, RHB may be more suitable than biochar produced from other types of agricultural waste for adsorbing heavy metals in water.

However, RHB has a poor adsorption capacity for specific metal ions in a polymetallic solution [13,117]. A study investigated the adsorption effectiveness of 21 types of biochar in polymetallic solutions. Yak dung biochar demonstrated the best Cu^2+^ adsorption, followed by cotton stem biochar, whereas RHB performed the worst. Yak dung biochar also demonstrated the best Pb^2+^ removal, and the worst was poplar wood biochar. However, RHB demonstrated the second-worst removal of Pb^2+^, which was only better than poplar wood biochar [118]. In addition, it was also found that the adsorption capacity of biochar was most inhibited in ternary metal fraction systems and most enhanced in unitary metal fraction systems [119]. Xu et al. [120] found that the adsorption of Zn^2+^ and Cd^2+^ by RHB was completely inhibited in a multi-metal solution system. Compared with the monometallic solution system, the adsorption capacities for Pb^2+^, Cu^2+^, Zn^2+^, and Cd^2+^ of dairy manure biochar were reduced by 2.00%, 21.1%, 40.9%, and 39.3%, respectively, in a polymetallic solution system. In contrast, the adsorption capacities of RHB were reduced by 38.4%, 42.7%, 92.3%, and 100%, respectively. Similarly, Zhang et al. [121] found that alfalfa biochar had higher adsorption capacities than RHB for both Cd^2+^ and Pb^2+^ coexisting in solution.

### 5.2. Adsorption of Organic Matter in Water

RHB mainly removes organic pollutants from water via adsorption and the electrostatic effect. The adsorption mechanism of organic pollutants by RHB is related to the pyrolysis temperature. RHB prepared at low pyrolysis temperatures (100 °C–300 °C) mainly adsorbs organic pollutants by partitioning. RHB prepared at higher pyrolysis temperatures (400 °C–700 °C) mainly adsorbs organic pollutants through chemisorption (e.g., π–π bonding, hydrogen bonding, and ligand bonding). This is because RHB produced at low temperatures has a high ash content. This causes inorganic minerals to occupy most of the adsorption sites on the RHB surface, rendering partitioning the main adsorption mechanism [122]. In contrast, RHB produced at a high temperature has a greater specific surface area, porosity, and aromaticity [123], rendering chemisorption the main adsorption mechanism.

The magnitude of the electrostatic effect is related to the pH value and zero point charge (pH_pzc_). In general, the RHB surface is positively charged when the solution pH is less than pH_pzc_. This inhibits the adsorption of cations in solution by RHB because of electrostatic repulsion. When solution pH > pH_pzc_, the RHB surface is deprotonated, rendering it negatively charged. This enhances the adsorption capacity of cations in solution because of electrostatic attraction [124,125]. Lingamdinne et al. [126] experimentally found that RHB adsorbed trinitrotoluene (TNT) and cyclotrimethylenetrinitramine (RDX) mainly via electrostatic interaction. However, RHB had a higher adsorption capacity for TNT than for RDX, which they attributed to TNT being an aromatic compound. This allows RHB to additionally adsorb TNT via chemisorption through π–π bonds and nitrogen–oxygen bonds. In contrast, RHB can only adsorb RDX via chemisorption through nitrogen–oxygen bonds.

## 6. Rice Husk Biochar as a Soil Conditioner

RHB is widely used for soil remediation and soil improvement. RHB not only improves the soil environment, promotes plant development, and increases yield but also passivates heavy metals and reduces their accumulation in plants. RHB can also increase the number and activity of soil microorganisms and restore a healthy ecological environment. However, adding RHB can also have negative effects that degrade the soil environment.

### 6.1. Effect on Soil pH

Applying RHB increases the pH of the soil. This effect is particularly obvious in acidic soils [127]. This is because RHB is inherently alkaline. In addition, RHB contains mineral elements such as calcium, potassium, magnesium, sodium, and silicon that form carbonates or oxides during pyrolysis. This reduces the exchange acidity by reacting with H^+^ and Al^3+^ in acidic soils, which further increases the soil pH [128]. Oladele [129] added RHB to soil and found that, after 3 years, the soil pH increased from 4.90 to 6.84 at a depth of 0–10 cm and from 5.12 to 6.62 at a depth of 10–20 cm for an average increase of 28%.

### 6.2. Effect on Soil Cation Exchange

Adding RHB generally increases soil cation exchange [130]. RHB increases ion exchange sites in soil because of its rich oxygen-containing functional groups, which increases the cation exchange capacity of soil. This subsequently increases the adsorption of cations by the soil. Oladele [129] reported that applying RHB increased the soil exchange of cations. Soil cation exchange is affected by the amount of RHB applied and the soil type [41]. This is because the different textures of the soil determine the function of the biochar once it was incorporated into the soil. Generally speaking, sandy soils have more grit, larger porosity and higher water holding capacity [131]. Ghorbani et al. [132] reported that applying 1% and 3% RHB to sandy soils increased the cation exchange by 20% and 30%, respectively. Moreover, applying 1% and 3% RHB to clay soils increased the cation exchange by 9% and 19%, respectively.

### 6.3. Effect on Soil Microorganisms

RHB has diverse effects on soil microorganisms that can be beneficial or detrimental. For example, the large specific surface area and porous structure of RHB provide shelter for soil microorganisms [133], and RHB provides nutrients for the growth of soil microorganisms. RHB also alters the microbial habitat by affecting the physicochemical properties of the soil, including the aeration conditions, water content, and pH. RHB induces changes in enzyme activity, which affects soil elemental cycles associated with microorganisms. RHB interrupts intraspecies and interspecies communication between microbial cells through the adsorption and hydrolysis of signaling molecules [134]. Finally, RHB enhances the adsorption and degradation of soil contaminants and reduces the biological effectiveness and toxicity of soil [135].

### 6.4. Effect on Heavy Metals in Soil

Research advances in heavy metal adsorption by biochar have steadily increased the mechanisms by which biochar removes heavy metals in soils. Sohi et al. [136] proposed three mechanisms (electrostatic interaction, ion exchange, and adsorption), Ahmad et al. [97] proposed five mechanisms (complexation, electrostatic interaction, ion exchange, physical adsorption, and precipitation), and Li et al. [137] proposed six mechanisms (complexation, electrostatic interaction, ion exchange, physical adsorption, precipitation, and reduction). Applying biochar to soil increases the electrostatic interactions between metal cations and activated functional groups of the soil by increasing the soil pH. Thus, in addition to direct interactions, biochar can indirectly reduce the uptake of heavy metals by plants by changing the soil properties [41].

Adding RHB can reduce the uptake of heavy metals by plants in composite soils. Derakhshan and Jung used RHB to remediate soil contaminated by multiple heavy metals. Compared with the control group, adding RHB reduced the concentrations of cadmium, copper, lead, and zinc in plant roots by 0.570, 0.6, 0.6, and 9.04 mg/kg, respectively. Bian et al. [138] found that adding RHB to contaminated soil reduced the cadmium and lead in cabbage leaves by 13% and 44%, respectively. Derakhshan and Jung [139] found that adding RHB to soil reduced the absorption of cadmium, copper, lead, and zinc in mustard plants by 79%, 13%, 87%, and 37%, respectively. However, Karam et al. [140] concluded that adding RHB increased the uptake of heavy metals by plants. This is because RHB has a high affinity for heavy metals, which prevents the migration and transformation of pollutants in the soil as well as the occurrence of severe soil contamination [141].

Adding RHB has been found to significantly increase the concentration of arsenic in soil. Zheng et al. [142] found that applying RHB to soil increased the arsenic content by 26%. Ibrahim et al. [143] found that adding RHB to soil significantly increased the arsenic content in the root, shoot, and leaf tissues of alfalfa grass. Zheng et al. [144] found that adding RHB increased the uptake of arsenic by wheat sprouts by 199%. These findings can be attributed to two main factors. First, arsenic is mainly present in soil in the form of anions such as AsO_3_^3−^ and AsO_4_^3−^. Applying RHB to soil increases the soil pH, which limits the ion exchange effect for anions [145]. Second, RHB is rich in substances such as silicon that compete with arsenic for adsorption sites, thereby increasing the mobility of arsenic [146].

### 6.5. Effect on Organic Pollutants in Soil

Applying RHB to soil can immobilize organic pollutants via adsorption. Adsorption mechanisms include partitioning, surface adsorption, and pore retention [147]. Partitioning is based on the principle of similar solubility for organic pollutants between the hydrophilic and hydrophobic phases. Surface adsorption is both physical and chemical. The physical process mainly involves van der Waals forces. The chemical process involves the formation of chemical bonds such as hydrogen, ionic dipole, coordination, and π–π bonds as well as intermolecular interactions. Pore retention involves RHB trapping of organic matter passing through its micropores. The organic matter is effectively isolated, which reduces the organic matter content of the soil [148].

### 6.6. Effect on Plant Growth

Applying RHB to soil generally improves crop yields. RHB directly provides crops with nutrients owing to its abundance of various minerals. RHB can also indirectly increase crop yield by improving the physicochemical properties of the soil, including the pH, cation exchange capacity, effective potassium, soil organic carbon, and soil bulk density [149]. Dong et al. [150] showed that applying RHB increased the rice yield from 6.66 ± 0.21 to 7.98 ± 0.55 t/ha. Similarly, Singh et al. [36] found that adding RHB increased the rice seed yield from 2.57 to 4.55 t/ha and the rice straw yield from 6.28 to 8.43 t/ha.

The high silica content of RHB facilitates the growth of crops than other biochar. Figure 3 shows that RHB is richer in amorphous silica than other biochar. Among the types of silicon, amorphous silica is most readily absorbed by crops [151]. In general, silicon in soil or biochar must be dissolved into silicic acid before it can be absorbed by crops [152]. When silicon from RHB enters the soil, it is absorbed by crop roots. At the same time, either –SiO(OH) is formed by interaction with hydrophilic compounds or silica gel is formed and precipitated upon hydrolysis. Then, the –SiO(OH) is absorbed and transported through the root cells to the stem via certain transporter proteins (Lsi1 and Lsi2), where it accumulates [153]. At this time, the silicon in the stem mainly exists in the form of monosilicic acids and, to a lesser extent, disilicate. Finally, the silicon (mainly monosilicic acids) in the stem is transferred to the branches via transpiration and is deposited as amorphous silica [154]. The Si in RHB absorbed by crops has the following effects. First, when plants do not receive the nutrients they need for growth, they can use Si [155]. Second, Si helps with crop growth, enhances disease resistance, promotes photosynthesis, and mitigates the production of reactive oxygen species, which can mitigate the harm caused by toxic and harmful substances [156]. Azhar et al. [157] found that applying RHB to cadmium-contaminated soil increased the chlorophyll content and photosynthesis rate by 12% and 122%, respectively. Third, Si reduces plant collapse and improves the mechanical strength of crop tissues and uprightness of rice plant stems and leaves [16]. Fourth, Si helps reduce crop water stress because it helps produce hemicellulose in the cell wall, which plays a crucial role in alleviating the water deficit in crops. Finally, Si reduces the uptake of heavy metals by crops. Si stimulates the production of certain substances in the crop root system that can chelate heavy metals and thus reduce their uptake [158].

## 7. Conclusions and Future Perspectives

Herein, we focused on the preparation and modification of RHB for the remediation of water and soil environments. Through an extensive examination of numerous studies on RHB production, we revealed that the parameters in the RHB preparation process have a considerable influence on the adsorption behavior of RHB. The adsorption capacity of unmodified biochar is limited. Generally, the specific surface area and functional groups of RHB can be enhanced through physical, chemical, and biological modification. RHB can remove pollutants from the environment through adsorption, ion exchange, electrostatic interaction, complexation, and cation exchange. However, RHB tends to be poor at adsorbing heavy metals when multiple heavy metals coexist in an environment. Moreover, RHB is an effective soil conditioner because it not only adsorbs heavy metals and organic matter but also improves crop growth by improving soil conditions and releasing beneficial elements (such as: silicon, potassium, calcium and magnesium). However, the pollutant adsorption capacity and remediation effectiveness of RHB can be affected by various factors in complex environments. This adds uncertainty to the results reported in the literature, and the mechanisms by which the effectiveness of RHB is affected require further investigation. Figure 4 shows the future perspectives of RHB.
(1)There is a necessity to strengthen the risk assessment and toxicology experiments on RHB as it is an adsorbent material with potential health hazards [161].(2)In the study of new RHB modification methods, the adsorption performance and secondary pollution characteristics of RHB must be comprehensively evaluated to design a nonpolluting modification method.(3)The natural environment often contains many different types of pollutants. Therefore, future research should focus on the comprehensive study of the sorption mechanism of RHB on one or several specific pollutants under coexistence of different types of pollutants (e.g., organic–inorganic composite pollutants and multiple heavy metal solutions).(4)To better utilize RHB in soil improvement, it is necessary to conduct pot trial studies to evaluate interactions between soil microorganisms and RHB, such as antagonistic and synergistic effects [162].(5)Long-term and regional field trials are required to study the value of RHB in agricultural applications to provide an accurate assessment of the use of RHB in this field.

**Figure 4 plants-12-01524-f004:**
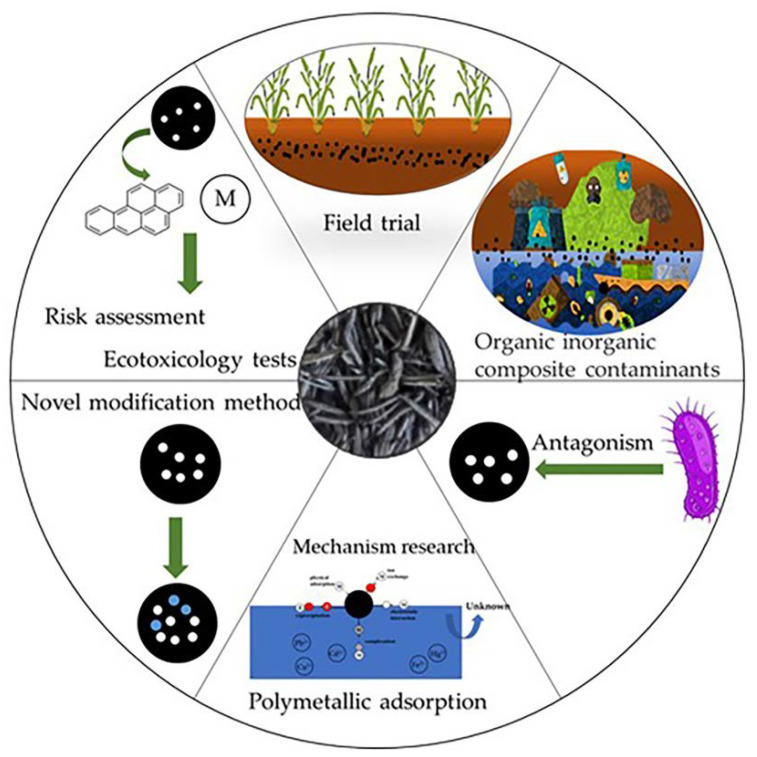
Future perspectives of rice husk biochar.

## Figures and Tables

**Figure 1 plants-12-01524-f001:**
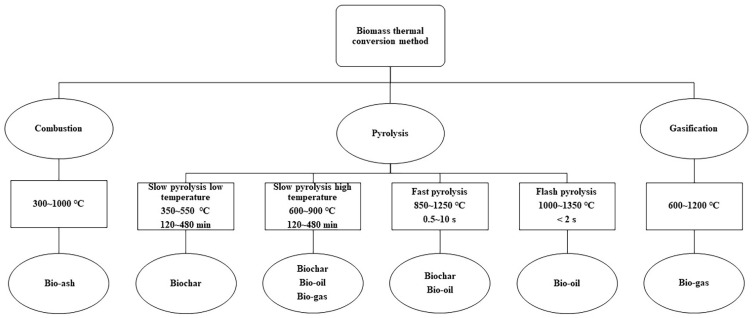
Thermochemical conversion modes of biomass fuels and their products [44,46].

**Figure 2 plants-12-01524-f002:**
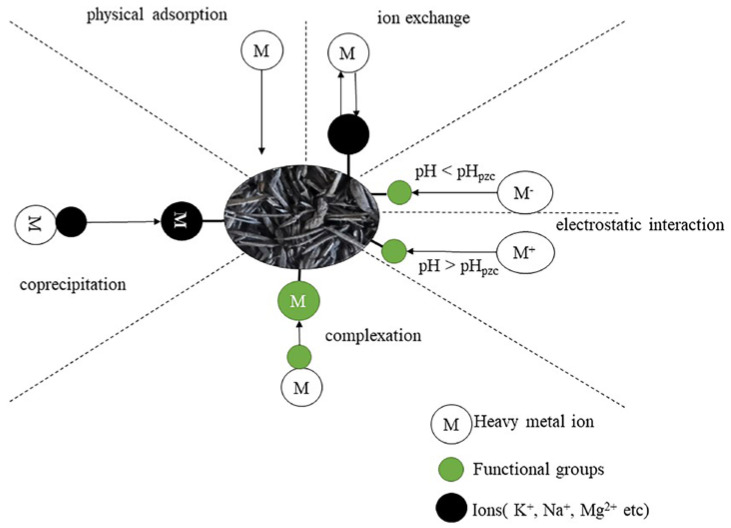
Removal mechanisms for rice husk biochar.

**Figure 3 plants-12-01524-f003:**
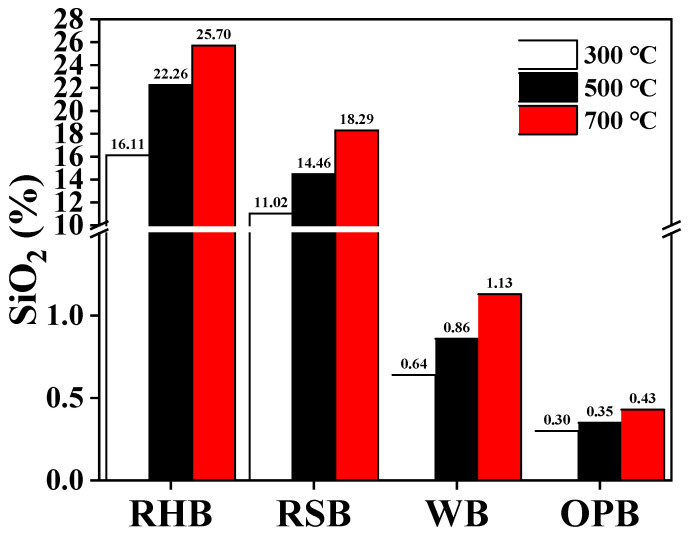
Silica contents of rice husk biochar and other biochar [152,159,160].

**Table 1 plants-12-01524-t001:** Characteristics and partial elemental composition of rice husk biochar.

Al (mg/kg)	Ca (mg/kg)	Si (mg/kg)	Mg (mg/kg)	As (mg/kg)	Reference
	0.02%		0.08%	0.55	Chatzimichailidou et al., 2023 [33]
189.00	691.00	11%	357.00		Samsuri et al., 2014 [34]
	0.81	35%	0.50		Severo et al., 2020 [35]
	225	168	184		Singh et al., 2018 [36]
92–543		66–199	162–658	1.79–2.50	Shackley et al., 2012 [37]
	8.24 cmol/kg	36.23	6.29 cmol/kg		Adebajo et al., 2022 [38]
	220	171	182		Varela et al., 2013 [39]
793	2245	166,338	900	3	Prakongkep et al., 2013 [40]
500	7804	149,449	1840		
212	1340	193,748	1683	3	

**Table 2 plants-12-01524-t002:** Characteristics of rice husk biochar at different pyrolysis temperatures.

Temperature (°C)	pH	H/C	O/C	SSA (m^2^/kg)	Reference
250–300	7.4	0.79	0.22		Abrishamkesh et al., 2015 [53]
300	5.7	0.09	0.42	8.0	Shen et al., 2021 [55]
500	9.8	0.04	0.14	25.0	
700	10.8	0.02	0.06	195	
300	8.65	1.192	0.264	6.54	Abbas et al., 2018 [56]
400	10.28	0.717	0.134	12.50	
500	11.36	0.578	0.067	20.11	
600	12.19	0.416	0.034	22.47	
300	7.47	0.89	0.61	2.57	Wei et al., 2017 [57]
500	10.47	0.42	0.53	18.4	
750	10.51	0.0199	0.679	53.08	
300	7.1	0.0695	0.3145	0.632	Shi et al., 2019 [58]
500	9.5	0.0459	0.1334	45.274	
700	9.8	0.0236	0.0848	193.149	
400		0.91	0.12	4.589	Liao et al., 2022 [59]
600		0.54	0.06	34.782	
350	6.41	1.10	0.52	11.61	Pariyar et al., 2020 [60]
450	6.92	0.91	0.30	18.58	
550	7.89	0.65	0.17	248.99	
650	7.97	0.58	0.11	280.97	
600	9.7	0.11	0.15	179.0	Pratiwi et al., 2016 [61]
300		0.0745	0.462	1.39	Yi et al., 2016 [62]
600		0.0396	0.201	168.0	
300	6.24	0.75	0.38	68.77	Mayakaduwa et al., 2017 [63]
500	7.17	0.51	0.37	169.81	
700	9.87	0.32	0.12	236.74	
700	10.72	0.47	0.24	242.53	Huang et al., 2020 [64]

**Table 3 plants-12-01524-t003:** Physical and chemical properties of RHB, corncob biochar, and pomelo peel biochar.

Biomass	SSA (m^2^/g)	H/C	O/C	Ash (%)	Reference
Rice husk	292.595	0.05	0.35	66.56	Jia et al., 2018 [104]
Rice husk	118.2		0.422	35.4	Severo et al., 2020 [35]
Rice husk	193.14	0.023	0.084	54.0	Shi et al., 2019 [58]
Rice husk	280.97	0.58	0.11		Pariyar et al., 2020 [60]
Rice husk	181.9	0.021	0.052	38.26	Wang et al., 2020 [105]
Corn cob	180.1	0.15	0.60		Liu et al., 2014 [106]
Corn cob	655.80	0.133	1.042	1.25	Suwunwong et al., 2020 [107]
Corn cob	14.589	0.136	0.772	8.30	Liao et al., 2022 [59]
Corn cob		0.37	0.14	4.0	Jing et al., 2018 [108]
Corn cob	10.38	0.025	0.112	5.25	Pipíška et al., 2022 [109]
sawn wood	243.1	0.08	0.29		Liu et al., 2014 [106]
sawn wood	32.8	0.35	0.11		Wan et al., 2016 [110]
sawn wood	2.946	0.08	0.71	1.2	Xu et al., 2019 [111]
sawn wood	86.59	0.147	1.205	1.42	Cheng et al., 2021 [112]
Wheat straw	0.67	0.072	0.286	118 g/kg	Mierzwa-Hersztek et al., 2020 [113]
Wheat straw	20.38	0.430	0.156	22.5	Manna et al., 2020 [114]
Wheat straw	58.38	0.040	0.443		Rajabi et al., 2021 [115]
Wheat straw	2.94	0.73	0.19	16.12	Chen et al., 2020 [116]

## Data Availability

Not applicable.

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
