# Peer review of "(untitled)"

_plants, 2023, doi:10.3390/plants12071524_

Round 1
Reviewer 1 Report
The manuscript entitled " Review on rice husk biochar as an adsorbent for soil and water remediation" presents an interesting research work because rice husk biochar has been found to be an effective adsorbent for soil and water remediation. Rice husk biochar has a high surface area and porosity, which makes it an ideal adsorbent for soil and water remediation. Additionally, it is non-toxic and biodegradable, making it a safe and eco-friendly option for environmental remediation. Overall, the manuscript has been well organized and clearly written. The authors are requested to consider following minor comments to improve their manuscript:
1. The abstract of the study is poorly written. An abstract can help readers decide whether or not they want to read the full paper, and can help them find relevant information more quickly. Additionally, an abstract can help increase the visibility of a study, as it can be indexed by search engines and databases.
2. Please mention common functional groups present on the surface of the biochar? What are the most effective methods for heavy metal remediation using biochar? How does biochar compare to other methods of heavy metal remediation?
3. What are the advantages and disadvantages of using biochar for heavy metal remediation? What are the environmental impacts of using biochar for heavy metal remediation?
4. What is the most common purpose of biochar’s surface modification? How does it works and how much it is cost effective? Make your introduction more attractive by adding more recent studies.
5. Please include a paragraph about how pyrolysis temperature effects on surface area, porosity, functional groups and adsorption capacity of biochar for pollutants.
6. The references below may of interest to you to further your discussion for a more expanded enrichment of information. Ex. J.Q. Zeng, et al, Pollution simulation and remediation strategy of a zinc smelting site based on multi-source information, Journal of Hazardous Materials 433 (2022) 128774; W.S. Ke, et al, Geochemical partitioning and spatial distribution of heavy metals in soils contaminated by lead smelting, Environmental Pollution 307 (2022) 119486.
Author Response
- The abstract of the study is poorly written. An abstract can help readers decide whether or not they want to read the full paper, and can help them find relevant information more quickly. Additionally, an abstract can help increase the visibility of a study, as it can be indexed by search engines and databases.
Reply: Thank you for your recognition of the work and contents of this manuscript. We have carefully rewritten the abstract in line 12-22 in response to your comments.
- Please mention common functional groups present on the surface of the biochar? What are the most effective methods for heavy metal remediation using biochar? How does biochar compare to other methods of heavy metal remediation?
Reply: Thank you for your suggestions! Your suggestions are very helpful, so we revised the content as you suggested.
(1) We have added in line 79-83 related to the common functional groups on the surface of biochar.
(2) In line 65-67, we add some common ways to improve the remediation of heavy metals by biochar. In addition, our literature review suggests that biochar can be used in combination with other technologies for heavy metal remediation, as described in line 67-70.
(3) In line 41-50, we added several common methods of heavy metal remediation and compared them with biochar
- What are the advantages and disadvantages of using biochar for heavy metal remediation? What are the environmental impacts of using biochar for heavy metal remediation?
Reply: Thank you for your suggestion!
(1) In line 55-60, we compared the advantages of biochar with some common materials used for heavy metal remediation, such as: activated carbon, fly ash, and clay minerals. In addition, in line 60-65, we added the disadvantages of using biochar for heavy metal remediation.
(2) The use of biochar for heavy metal remediation can have both beneficial and detrimental effects on the environment. We added the favorable effects in line 76-79,85-87 and the unfavorable effects in line 60-65.
- What is the most common purpose of biochar’s surface modification? How does it works and how much it is cost effective? Make your introduction more attractive by adding more recent studies.
Reply:
Thank you for your suggestion. We mentioned the purpose and work of biochar’s surface modification. and we added the cost of biochar modification in line 201-211 to make the introduction more attractive and interesting.
- Please include a paragraph about how pyrolysis temperature effects on surface area, porosity, functional groups and adsorption capacity of biochar for pollutants.
Reply: Thank you for your suggestions! In line 157-181, we rewrote the effect of pyrolysis temperature on the physicochemical properties of biochar.
- The references below may of interest to you to further your discussion for a more expanded enrichment of information. Ex. J.Q. Zeng, et al, Pollution simulation and remediation strategy of a zinc smelting site based on multi-source information, Journal of Hazardous Materials 433 (2022) 128774; W.S. Ke, et al, Geochemical partitioning and spatial distribution of heavy metals in soils contaminated by lead smelting, Environmental Pollution 307 (2022) 119486.
Reply: Thank you for your recommendation. We have reviewed relevant references and added the contents about spatial distribution and mobility of heavy metals (See line 37-40)
Reviewer 2 Report
The manuscript is well written but needs some attention. Rice husk biochar is a potential health hazard material as it contains both amorphous and crystalline silica and I suggest author should write a paragraph on this as producer and users know the potential human health hazard and environmental issue from rice husk biochar itself may arise.
Author Response
Thank you for your suggestion! The potential harm problem of rice husk biochar really deserves our attention. In line 95-110, we added a paragraph to describe this problem. And we also added a table about the characteristics of rice husk biochar (See Table 1).
Reviewer 3 Report
Aiming the high value utilization of the rice husk, authors summarized the preparation and modification of rice husk biochar for the remediation of water and soil environments. Detailed description and comparison have been performed in this review article. In general, it is a good topic to be deserved a review article. The manuscript is well organized. However, there are still some issues to be addressed. A moderate revision is suggested before its acceptance.
1. As a review article, comprehensive article collection, analysis, and discussion are necessary. More recent and highly relevant articles should be added into this review article.
2. The experimental evaluation of rice husk ash for mortars should be introduced and compared in the manuscript: Experimental evaluation of rice husk ash for applications in geopolymer mortars
3. At the end of introduction, it is better to scheme one figure to summarize the whole review contents
4. More figures should be added to enrich the content of this review article, and also better for the understanding to readers.
5. The contents from the template should be deleted.
6. More introduction on the formation, structure, properties and application of biochar should be provided with supporting articles: Biochar derived from non-customized matamba fruit shell as an adsorbent for wastewater treatment; Synthesis and Application of Granular Activated Carbon from Biomass Waste Materials for Water Treatment: A Review; Production of solid fuels by hydrothermal treatment of wastes of biomass, plastic, and biomass/plastic mixtures: A review; etc.
7. In section 7, more perspectives should be added on the challenges and possible solutions to guide the future studies. In addition, one figure is suggested to show the challenges and possible solutions.
8. There are still some typos and grammar issues in the manuscript to be corrected.
9. All the references should be rechecked to make sure all the information is provided, such as the volume and pages.
Author Response
- As a review article, comprehensive article collection, analysis, and discussion are necessary. More recent and highly relevant articles should be added into this review article.
Reply: Thank you for your recognition of the work and contents of this manuscript. We have reviewed the relevant study and supplemented some contents to make the article more attractive and interesting. In line 55-65, we compared the advantages and disadvantages of biochar with some common materials used for heavy metal remediation. In line 79-83, we added the common functional groups on the surface of biochar. In line 95-110, we added a paragraph to describe the potential harm problem of rice husk biochar. In line 157-181, we rewrote the effect of pyrolysis temperature on the physicochemical properties of biochar.
- The experimental evaluation of rice husk ash for mortars should be introduced and compared in the manuscript: Experimental evaluation of rice husk ash for applications in geopolymer mortars
Reply: Thank you for your suggestion. In line 87-94, we have added some descriptions of rice husk ash as filler on the compressive and sulfate resistance of mortar.
- At the end of introduction, it is better to scheme one figure to summarize the whole review contents
Reply: Thank you for your suggestion. Your suggestion is of great help to us. We have added a figure (Graphical abstract) to summarize the whole review contents to make the contents more intuitive in the end of the abstract.
- More figures should be added to enrich the contents of this review article, and also better for the understanding to readers.
Reply: Thank you for your suggestion. We added some tables and figures to enrich the contents of this review article. In line 110, we add a table (Table 1) to describe the characteristics and partial elemental composition of rice husk biochar. In line 518, we also added a table about the future perspectives of rice husk biochar.
- The contents from the template should be deleted.
Reply: Thank you for your suggestion. We have deleted the contents from the template and revised these paragraphs in line 503-516.
- More introduction on the formation, structure, properties and application of biochar should be provided with supporting articles: Biochar derived from non-customized matamba fruit shell as an adsorbent for wastewater treatment; Synthesis and Application of Granular Activated Carbon from Biomass Waste Materials for Water Treatment: A Review; Production of solid fuels by hydrothermal treatment of wastes of biomass, plastic, and biomass/plastic mixtures: A review; etc.
Reply: Thank you for your recommendation.
(1) We have reviewed relevant references and added the contents about compared the advantages of biochar with activated carbon used for heavy metal remediation (See line 55-57).
(2) In line 51-54, we added the introduction about the formation of rice husk biochar.
(3) We added some contents about the structureand properties of rice husk biochar (See line 48-50,159-181).
(4) About the application of rice husk biochar, we added the effect of rice husk ash as filler on the compressive properties and sulfate resistance of mortar (See line 87-94).
- In section 7, more perspectives should be added on the challenges and possible solutions to guide the future studies. In addition, one figure is suggested to show the challenges and possible solutions.
Reply: Thank you for your suggestion. In line 489-516, we have added more future perspectives. And we also added a figure (Figure 4) about the future perspectives of rice husk biochar.
- There are still some typos and grammar issues in the manuscript to be corrected.
Reply: Thank you for your suggestion. We have carefully checked and corrected the typos and grammar issues in our manuscript.
(1) In line 39, “a” has been deleted.
(2) In line 44, “need” has been changed into “demand”
(3) In the whole manuscript, “ mg⸳kg-1” has been changed into “mg/kg”, The font of temperature has been corrected, “figure” has been changed into “Figure” and “table” has been changed into “Table”.
(4) In line 304, “is” has been changed into “are”.
(5) In line 363-364, “Nitrogen-oxygen double bond” has been changed into “Nitrogen-oxygen bond”.
(6) In line 432, “solidify” has been changed into “immobilize”.
- All the references should be rechecked to make sure all the information is provided, such as the volume and pages.
Reply: Thank you for your suggestion. We have corrected reference style in line 550, 615, 618, 667, 701, 703, 715, 848.
Round 2
Reviewer 2 Report
All good
Author Response
Thanks for your help!